# *Paracoccidioides* Species Circulating in the Endemic Area of Rio de Janeiro, Brazil: Updates into Their Genetic Diversity

**DOI:** 10.3390/jof11020134

**Published:** 2025-02-10

**Authors:** Beatriz da Silva Motta, Fernando Almeida-Silva, Marcus de Melo Teixeira, Andréa Reis Bernardes-Engemann, Rodrigo Almeida-Paes, Priscila Marques de Macedo, Rosely Maria Zancopé-Oliveira

**Affiliations:** 1Fundação Oswaldo Cruz, Instituto Nacional de Infectologia Evandro Chagas, Laboratório de Micologia, Rio de Janeiro 21040-360, Brazil; beatriz.motta@ini.fiocruz.br (B.d.S.M.); fernando.almeida@ini.fiocruz.br (F.A.-S.); andreaengemann@yahoo.com.br (A.R.B.-E.); rodrigo.paes@ini.fiocruz.br (R.A.-P.); 2School of Medicine, University of Brasilia, Brasilia 70910-900, Brazil; marcus.teixeira@gmail.com; 3Fundação Oswaldo Cruz, Instituto Nacional de Infectologia Evandro Chagas, Laboratório de Pesquisa Clínica em Dermatologia Infecciosa, Rio de Janeiro 21040-360, Brazil; priscila.marques@ini.fiocruz.br

**Keywords:** paracoccidioidomycosis, *Paracoccidioides*, *P. americana*, DNA barcoding

## Abstract

Paracoccidiodomycosis (PCM) is the most important systemic mycosis in Brazil, and is usually associated with rural work. PCM is caused by inhalation of infective propagules of thermodimorphic fungi from the genus *Paracoccidioides*. In the past, it was believed that *Paracoccidioides brasiliensis* was the single species responsible for PCM cases. However, recent advances in molecular methods allowed the description of several new species, using phylogenetic concordance as the gold standard. Aside from *P. brasiliensis sensu stricto*, *Paracoccidioides americana* is also endemic in Rio de Janeiro state, Brazil. This study aimed to evaluate intraspecific genetic variability of *Paracoccidioides* isolates from patients diagnosed with PCM at a reference center for endemic mycoses in Rio de Janeiro state, from 2015 to 2021. Among the sixteen retrieved isolates, three (18.75%) were identified as *P. americana* and thirteen (81.25%) as *P. brasiliensis sensu stricto*. No intraspecific genetic variation was observed by the M-13 primer in *P. americana* isolates from this geographic region. However, *P. brasiliensis* sensu stricto isolates were clustered into two distinct molecular profiles, despite being grouped in a single clade in the phylogenetic tree after partial sequencing of *arf* and *gp43* genes. The results suggest a single *P. americana* lineage and two *P. brasiliensis* populations causing PCM in Rio de Janeiro, Brazil.

## 1. Introduction

Paracoccidioidomycosis (PCM) is the most important systemic mycosis in Brazil with high levels of morbidity and mortality. The disease is traditionally associated with rural work, but recent modifications on PCM epidemiology have been described, with the emergence of urban cases of the disease in the Southeastern Brazilian state of Rio de Janeiro [1,2]. In 2017, an acute PCM outbreak was reported after a highway construction in Rio de Janeiro state and the agents identified were *Paracoccidioides brasiliensis sensu stricto* and *Paracoccidioides americana*, suggesting the sympatry of these two species [3].

Furthermore, PCM geographic expansion has been observed in the Brazilian territory, with increasing cases in the Northern region due to deforestation and land-use dynamics for agriculture commodities [4]. The main risk factors for PCM infection include middle-aged men, smokers, and working in a rural environment [5,6].

The infection is acquired through inhalation of conidia and other infective propagules of the thermodimorphic fungus *Paracoccidioides* spp. After reach the pulmonary alveoli, they will convert to the yeast phase and are subsequently phagocytosed by cells of the mononuclear phagocytic system with the formation of the primary pulmonary ganglion complex [7,8]. Depending on the host immune status and the inhaled fungal burden and the virulence of the infecting strain, there will be a complete resolution, infection, or lymphohematogenous dissemination and progression to clinical disease [9]. Therefore, PCM is clinically classified as infection, disease (acute or chronic forms), and residual forms or sequelae [10]. PCM infection occurs in individuals exposed to environmental sources of *Paracoccidioides*, presenting effective cellular immune response, without clinical manifestations [5]. Chronic PCM is the most frequent clinical form, occurring after a long latency period, in adult patients intensely exposed to soil management activities and affecting mainly the lungs, oral mucosa, and pharynx/larynx. Acute PCM usually occurs in genetically predisposed young patients presenting a humoral response unable to control fungal infection, then quickly evolving with invasive disease characterized by enlarged lymph nodes with liver and spleen involvement [5,9].

Regarding PCM case definition criteria, a suspected case consists of clinical and epidemiological features, probable cases being defined by the presence of anti-*Paracoccidioides* circulating serum antibodies, while confirmed cases present the identification of the fungal agent in fresh-tissue and histopathological examinations, or culture. The gold standard diagnostic method is culture, and fungal thermal dimorphism is demonstrated through conversion from filamentous to the yeast phase [11,12]. Serology plays a key role in the presumptive diagnosis of classic PCM by detecting circulating antigens or antibodies and is suitable for patient follow-up [13].

In the past, it was believed that *P. brasiliensis* was the single species responsible for all PCM cases [14]. However, advances in molecular identification methods allowed the description of a complex of species, including three phylogenetic species: *P. brasiliensis* S1, PS2, and PS3 [15,16]. Later, another genetic variant was described as Pb01-like, which, due to its higher genetic and morphologic diversity, was named as *Paracoccidioides lutzii* [17,18]. The phylogenetic species PS4 was the fifth and last one described [19].

More recently, whole genome sequencing (WGS) analyses described two clades within the phylogenetic species S1, which were named S1a and S1b [20]. Thus, according to the Brazilian consensus on PCM, *P. brasiliensis*, which is composed of a complex of five phylogenetic species (S1a, S1b, PS2, PS3, and PS4), and *P. lutzii* are the species that currently cause PCM [10]. These phylogenetic species present ecological, epidemiological, and antigenic expression variations, which thus could cause changes in laboratory diagnosis and therapeutic response [18,19].

In 2017, a publication evaluated the phylogenetic profile of nuclear and mitochondrial DNA as well as morphological characteristics of the four phylogenetic species of *P. brasiliensis*, and proposed considering them as taxonomic species, namely, *P. brasiliensis sensu stricto* (S1), *P. americana* (PS2), *Paracoccidioides restrepiensis* (PS3), and *Paracoccidioides venezuelensis* (PS4) [21].

A more recent publication described two new species of *Paracoccidioides*, namely, *Paracoccidioides ceti* and *Paracoccidioides lobogeorgii*. *P. ceti* can be found in dolphins that circulate in oceans with contact with rivers in Latin America, while the latter is found in individuals living in the Amazon Basin and Latin American countries [22,23]. These two species, considered uncultivable, can cause PCM loboi and PCM ceti, which are subcutaneous mycoses and may manifest as keloid-like lesions in humans and dolphins, respectively [12].

Most molecular techniques above described use DNA extracted from fungal cultures and were developed before the recognition of the new species of the genus *Paracoccidioides*, thus allowing partial characterizations of the isolates [11,14,21,24,25,26]. Sanger sequencing is the gold standard to identify *Paracoccidioides* spp., but this method presents some limitations [11,27].

Despite the knowledge of new *Paracoccidioides* species, routine diagnosis of PCM is currently based on conventional methods, through identification of the fungal agent in clinical samples and the detection of specific antibodies or antigens in serological tests, while molecular methods have not been implemented in the routine diagnosis of PCM yet, being used as an alternative test [28,29]. Furthermore, identifying the fungal species is not required for the clinical and therapeutic management of PCM cases so far [11].

In 2018, the clinical features and molecular identification of clinical isolates from a cohort of patients with PCM in the state of Rio de Janeiro were described for the first time, with *P. brasiliensis sensu stricto* and *P. americana* endemic species detected in this region. Until then, *P. americana* was considered a rare species, scarcely explored [30].

Considering the genetic profile occurring in this endemic area, this study aimed to perform a more robust molecular characterization of the species *P. americana* isolated from clinical samples of patients diagnosed with PCM at a reference center for this mycosis in the state of Rio de Janeiro, from 2015 to 2021, thus evaluating intraspecific variability of these fungal isolates and providing an update of the *Paracoccidioides* species present in this important endemic area of Brazil.

## 2. Materials and Methods

### 2.1. Fungal Isolates

Fungal isolates (*n* = 36) identified morphologically as *Paracoccidioides* spp. in the routine diagnosis from clinical samples of patients diagnosed with PCM at the Laboratório de Referência Nacional em Micoses Sistêmicas, Instituto Nacional de Infectologia Evandro Chagas, Fundação Oswaldo Cruz (INI/Fiocruz), from 2015 to 2021, were included in this study. These clinical isolates were stored under sterile mineral oil at 25 °C, without prior molecular identification, and recovered using two culture media. Initially, a fragment of the stored colony was seeded onto Potato Dextrose Agar (PDA) medium (Becton, Dickinson and Company, Sparks, MD, USA) at 25 °C to obtain the filamentous form. After approximately 30 days, the recovered colonies were subcultured on Fava-Netto Agar medium [31], being converted to the yeast-like form at 35 °C. After morphological conversion, colonies were periodically subcultured on Fava-Netto Agar medium every 7 to 14 days for maintenance and preservation of isolates.

### 2.2. Phenotypic Identification of Isolates Grown on PDA and Fava-Netto Culture Media

All fungal isolates were grown on PDA (Becton, Dickinson and Company, Sparks, MD, USA) and Fava-Netto culture media, with incubation at 25 °C and 37 °C, respectively, to obtain their filamentous and yeast forms, which were identified by their macro and micromorphological characteristics, according to the description made by Lacaz and collaborators [32].

### 2.3. DNA Extraction

DNA extraction was performed from the yeast-like phase using a previous published protocol [33] with minor modifications in the lysis protocol. Briefly, it consisted of collecting the yeast fungal mass obtained in Fava-Netto medium and transferring it to a 2 mL tube, in which a 500 µL lysis solution consisting of 100 mM Tris pH 8.50 mM EDTA, 1% SDS (TES lysis buffer), and 0.5 mm zirconia beads (BioSpec Products, Bartlesville, OK, USA) were added. Subsequently the solution was vigorously mixed in a mini-bead beater (BioSpec Products, Bartlesville, OK, USA) for three cycles of 30 s with 4000 oscillation/minute, aiming to break down the yeast-like cells. The following steps were performed according to the prior protocol description. Finally, the obtained DNA was suspended on 50 µL of ultra-pure distilled water (Thermo Fisher Scientific, Waltham, MA, USA). DNA quantification and purity were analyzed by spectrophotometry using the Eppendorf BioPhotometer (Eppendorf AG, Hamburg, Germany).

### 2.4. Molecular Identification

The molecular identification of the recovered *Paracoccidioides* spp. isolates was performed through the partial sequencing of *arf* and *gp43* genes as previously described [34,35]. Briefly, the following primers *arf* forward 5′-TCTCATGGTTGGCCTCGATGCTGCC-3′, *arf* reverse 5′-GAGCCTCGACGACACGGTCACGATC-3′, *gp43* exon 2 forward 5′-CCAGGAGGCGTGCAGGTGTCCC-3′, and *gp43* exon 2 reverse 5′-GCCCCCTCCGTCTTCCATGTCC-3′ [15,18] were used for PCR reactions using the Platinum *Taq* DNA polymerase PCR Master Mix (2X) (Invitrogen, Waltham, MA, USA). Amplicons were purified using QIAquick PCR Purification Kit (QIAGEN, Hilden, Germany) according to the manufacturer’s instructions, being reconstituted with 50 µL of ultra-pure water.

The Sanger method [36] was performed to provide partial DNA sequencing at the DNA Sequencing Platform PDTIS/Fiocruz using the Applied Biosystems ABI Prism 3730 instrument (Applied Biosystem, Foster, CA, USA). The sequences obtained were examined using the Sequencher Software Package version 4.9 (GeneCodes Corporation, Ann Arbor, MI, USA) and aligned with MEGA software version 10 [37]. After this, the sequences generated were used for the phylogenetic comparison with reference sequences retrieved from the GenBank database, including the sequences previously described in the same study region [30]. A maximum likelihood tree was generated using 1000 bootstrap replicates in the IQ-TREE 2 software Version 2.2.6 [38]. Phylogenetic trees were constructed with the help of the FigTree software v 1.4.4. (http://tree.bio.ed.ac.uk/software/figtree/ (accessed on 3 March 2024)).

### 2.5. Intraspecific Genetic Variability

To access intraspecific genetic variability among isolates belonging to each species of *Paracoccidioides*, an M13 wild-type phage minisatellite-specific core sequence (5’-GAGGGTG GCGGTTCT-3′) was employed as a single primer in the PCR protocol [39]. According to Vassart and colleagues [40], this method using the phage M13 sequence detects hypervariable minisatellites in human and animal DNA. The PCR was performed with the same conditions previously established by Moreira, as follows: the amplification reactions were performed with Platinum *Taq* DNA polymerase PCR Master Mix (2X) (Invitrogen, Waltham, MA, USA), composed of a 50 µL mixture containing 75 ng of genomic DNA; reaction buffer 10X (dilution 1:10, 20 mM TrisHCl pH 8.4, 50 mM KCl) (Invitrogen, Waltham, MA, USA); 1.5 mM MgCl_2_ (Invitrogen, Waltham, MA, USA); 0.2 mM dNTP Mix (Invitrogen, Waltham, MA, USA); 3.0 mM magnesium acetate; 2.5 U Platinum Taq DNA polymerase (Invitrogen, Waltham, MA, USA); 30 ng primer; and ultra-pure distilled water (Thermo Fisher Scientific, Waltham, MA, USA). PCR was performed in a T100 Thermal Cycler (Bio-Rad, Hercules, CA, USA), and the program consisted of 35 cycles of 20 s of denaturation at 94 °C, 1 min of annealing at 40.4 °C, 20 s of extension at 72 °C, followed by a final extension for 10 min at 72 °C [41]. Amplification products were removed, concentrated to approximately 20 µL, separated by electrophoresis on 1.4% agarose gels (stained with ethidium bromide [final concentration, 0.5 g/mL]) in 0.5X TBE buffer at 60 V for 14 cm, and visualized under UV light. All visualized bands on the gels were counted, independent of the intensity, and the presence or absence of the amplified DNA bands was recorded [42]. The DNA from six clinical isolates of *P. americana* previously identified by our group [30] as well as isolates identified in this study were included in this test. We also incorporated the DNA from reference strains Pb03 (*P. americana*) and Pb18 (*P. brasiliensis sensu stricto*). Data analysis was based on the Dice similarity coefficients, being grouped using the unweighted pair group method with arithmetic mean (UPGMA) with the NTSYS-pc software (version 2.02h, Applied Biostatistic Inc., Foster, CA, USA).

The whole genome sequencing was another method performed to evaluate the intraspecific genetic variability of *P. americana* isolates. For this, the DNA from four clinical isolates of *P. americana* previously identified by our group [30], as well as DNA from *P. americana* (n = 3) isolated from patients with PCM from Rio de Janeiro state and treated at INI/Fiocruz during this study, were included in our analysis. Sequencing was realized with the Illumina novaseq 6000 platform at a minimum coverage of 100X for evolutionary analyses. Illumina reads were inspected for quality by the FastQC platform and adapters, and contaminating reads were removed using the BBDuk and BBmap modules from BBTools package (v 38.95). To identify SNPs, the reads of seven herein analyzed genomes and 70 additional sequence data from previously published *Paracoccidioides* spp. genomes [20,43] were mapped against the Pb01 reference genome using the BWAmem software (version BWA-0.7.16); .bam files were generated and SNPs were identified by GATK v4 HaplotypeCaller pipeline in haploid mode generating .vcf files. Finally, SNPs were filtered to remove those that have a coverage of less than 10x, allelic variation of less than 90%, or were identified in regions of gene duplication when compared with the reference genome. Phylogenomic and population genetic analyses were realized with SNPs obtained in the IQ-TREE 2 software Version 2.2.6 [38].

## 3. Results

### 3.1. Phenotypic Identification

Fungal isolates grown at 25 °C presented macromorphologically as flat, wrinkled, and folded glabrous colonies, suede-like in texture, compared to a “burst popcorn”, white to brownish with a tan or brown reverse. Hyaline and septate hyphae and chlamydospores were microscopically visualized (Figure 1A,B). The yeast-like form (37 °C) produced a white to tan colony with “brain-like” aspect in the macromorphology. In microscopy, large, round, yeast-like cells were observed. Multiple budding occurred, being thick-walled and birefringent cells similar to a “steering wheel” (Figure 1C,D).

### 3.2. Molecular Identification by Sequencing

Sixteen isolates of *Paracoccidioides* spp. were recovered at the Laboratório de Referência Nacional em Micoses Sistêmicas, INI/Fiocruz, in the period of the study, representing a recovery rate of 44.44%. Among them, 13 (81.25%) were identified as *P. brasiliensis sensu stricto* and 3 (18.75%) as *P. americana* by the Sanger method [36]. The acute form of PCM prevailed in patients affected by *P. brasiliensis* (n = 8/13; 61.54%), while the chronic form was attributed to all patients infected with *P. americana*. Regarding the probable place of geographic origin of the clinical isolates, that is, the most likely place where patients became infected by the fungus, *P. brasiliensis* isolates were mostly from Rio de Janeiro state (*n* = 12), followed by one isolate from Minas Gerais (Southern Brazil). All *P. americana* isolates were recovered from patients that were born and live in the Rio de Janeiro state. Table 1 depicts the main clinical-epidemiological characteristics of the fungal isolates included in this study.

Figure 2 represents the phylogenetic tree built from the analysis of the sequences from this study, as well as sequences from the reference strains. Concerning the phylogenic analysis of the *Paracoccidioides* spp. included in this study, the sequences generated for both genes herein studied showed more than 99% identity when compared with the reference strains deposited in GenBank (KU645891.1 and KU645890.1 for *P. americana*; KU042925.1 and KU042924.1 for *P. brasiliensis sensu stricto*).

#### Relatedness Among Rio de Janeiro Sequences and Other Sequences from a Public Data Bank

We subjected the *arf* and *gp43* sequences that we obtained above to BLAST analysis with data sets for *P. americana* and *P. brasiliensis* sequences in the NCBI public databank in order to assess the relationship between our isolates included in this study and *Paracoccidioides* spp. isolates from other regions. We selected sequences that provided 100% query coverage and exhibited maximum similarity. An analysis of the evolutionary relationship of the 181 taxa (Rio de Janeiro isolates plus isolates from other regions) was conducted using MEGA software version 10. The phylogenetic tree (Figure 2) contained five distinct major clades comprising the five phylogenetic *Paracoccidioides* species. Analysis of the identified data sets demonstrated that the sequences of *P. brasiliensis sensu stricto* were similarly very close among the isolates from other regions, and, as already demonstrated, our isolates presented 99% similarity with them. As previously observed, two distinct clades were observed within *P. americana*. Clade A is composed only of strains recovered in Rio de Janeiro, suggesting local adaptation, and was genetically apart from Clade B, which only includes isolates from the state of São Paulo. The phylogenetic analysis corroborated our previous identification, and no *Paracoccidioides* isolate included in this study showed identity with *P. restrepiensis*, *P. venezuelensis*, and *P. lutzii*.

### 3.3. Intraspecific Genetic Variability

#### 3.3.1. M13 Wild-Type Phage Minisatellite-Specific Core

In this study, the predominant species detected was *P. brasiliensis sensu stricto*, followed by *P. americana*. After evaluating the 10 samples of *P. americana* (6 previously identified by our group [30]; 3 identified in this study; and Pb03) and 14 of *P. brasiliensis sensu stricto* (13 identified in this study and Pb18), three reproducible PCR fingerprinting profiles (designated Clades 1 to 3) were obtained with primer M13, indicating that there was a considerable level of genetic diversity among the *Paracoccidioides* strains isolated from Rio de Janeiro, an important Brazilian endemic area. After obtaining the Dice similarity coefficients, the genetic relationships determined with the UPGMA were represented by a dendrogram (cophenetic correlation coefficient, 0.941) (Figure 3). *P. americana* isolates were grouped into a single clade (profile 1), without intraspecific genetic variation. The *P. brasiliensis sensu stricto* isolates were divided into two clades (profiles 2 and 3), showing a genetic diversity among isolates of this species, even belonging to the same geographic region in the state of Rio de Janeiro. Despite this, Clade 2 held most of the *P. brasiliensis sensu stricto* isolates evaluated (*n* = 11/14; 78.57%), considering ten clinical isolates and Pb18. The acute form of PCM was prevalent in Clade 2 (*n* = 8; 80%), while Clade 3 grouped only cases of the chronic form.

#### 3.3.2. Whole Genome Sequencing

Genetic differences were observed between seven isolates of *P. americana* studied (four previously identified by our group [30] and three identified in this study), indicating two distinct populations in the species of *P. americana*, according to phylogenetic evaluation (Figure 4A). The whole genome sequencing revealed that Rio de Janeiro isolates are more diverse than the population of others regions according to nucleotide diversity (π) and PCA analyses (Figure 4B,C).

## 4. Discussion

For several decades, PCM has been widely identified in Southeast Brazil, Colombia, and Venezuela, but, due to the dense expansion of agricultural frontiers in the Midwest and the Brazilian Amazon during the last 30 years, the epidemiology of this disease has undergone modifications. Therefore, it is fundamental to perform molecular epidemiological surveys in all of those regions [5]. In fact, Roberto and collaborators [44] stated the need for genetic surveillance in endemic areas in order to guarantee that molecular epidemiological studies are accurate, due to the epidemiological data recently reported for *Paracoccidioides* spp., with the high number of cases in different Latin American countries. Rio de Janeiro is the third state with the highest number of hospitalizations due to PCM in Brazil, and it has recently been facing important epidemiological modifications, such as the description of two endemic species in this state, *P. americana* and *P. brasiliensis sensu stricto* [1,30]. The geographic expansion of PCM cases makes epidemiological surveillance essential to determine which *Paracoccidioides* species are present in a given region. Furthermore, through epidemiological surveillance, it is possible to use antigen preparations that include antigens from the prevalent species for a more sensitive serological diagnosis. With a more reliable presumptive result, it is possible to initiate early and appropriate treatment for PCM. Therefore, it is necessary for continuous identification of which species of *Paracoccidioides* spp. are recurrently isolated in this endemic area for more accurate clinical decisions to be made in this endemic area.

Similarly to the findings of the previously mentioned study [30], we found *P. brasiliensis sensu stricto* and *P. americana* as the species currently circulating in the state of Rio de Janeiro, Brazil. These results, along with the findings regarding the origin of the isolates, may suggest that these species cause autochthonous cases, as most infected individuals are residents and developed risk activities for PCM in the state of Rio de Janeiro. Since 2016—the year of the first publication of a *P. americana* case in this state occurred in 2002—it has been possible to suggest that there are two species endemic in this region, sharing similar habitats and producing related clinical characteristics for decades in this endemic area [30,34,45].

The first autochthonous case of PCM caused by *P. americana* in the state of Rio de Janeiro refers to a patient who presented the chronic form of the disease [45]. In addition to this case, five other patients affected by this species were described in this region, all but one presenting the PCM chronic form. The patient with the acute form was diagnosed during the acute PCM outbreak described in this region [3,30]. In the current study, all patients infected by *P. americana* presented with the chronic PCM form, thus reinforcing a possible tendency of this species to induce chronic presentations in the infected patients.

We previously reported that M13 PCR fingerprinting was a useful tool for identifying molecular types of *H. capsulatum* in different geographic regions of Brazil [46]. However, to date, there are no studies evaluating the intraspecific genetic variability of *P. americana* isolates by this method, possibly due to the rarity of this species. However, this species is mostly found in the geographical region herein studied, and allowed the application of this discriminatory genetic test. In this study, we observed that the *P. americana* isolates exhibited the same molecular profile. It is noteworthy that the reference strain of *P. americana* (Pb03) does not come from Rio de Janeiro, but from São Paulo. Although isolates of *P. americana* from these two regions are grouped in different clades through *arf* and *gp43* sequencing, showing a possible geographical isolation in the process of species diversification, the reference strain from São Paulo presented, in the M13 fingerprinting, the same molecular profile of the strains from Rio de Janeiro, presenting hypervariable repetitive regions, called minisatellites, similar to those detected in the isolates of our endemic area. On the other hand, isolates of *P. brasiliensis sensu stricto* showed a notable intraspecific variability, indicating a higher level of genetic variation, which has been undergoing mutations over the years, perhaps due to the geographic expansion and epidemiological changes regarding the disease in recent years. Interestingly, those two clades of *P. brasiliensis* were composed exclusively or predominantly of isolates causing chronic or acute PCM, respectively, which may indicate genetic determinants of the major clinical forms of PCM. A larger study, with *P. brasiliensis* isolates from other endemic areas, is necessary to assess this hypothesis.

As well as M13 PCR fingerprinting, there are no studies evaluating the intraspecific genetic variability of *P. americana* isolates by whole genome sequencing analysis. According to phylogenetic evaluation, there are two distinct populations in the species of *P. americana* based on whole genome sequencing analysis; the PCA and neighboring tree unrooted of *P. americana* isolates evidenced the existence of two distinct populations of this species. Considering our results, the seven isolates of *P. americana* studied were clustered in Clade 1 of the maximum likelihood tree based on the SNPs collected, so both techniques evidenced that there is no intraspecific genetic variability between isolates of *P. americana* from Rio de Janeiro, and these belong to the same population. Therefore, the Rio de Janeiro population is more diverse than the population of other regions of Brazil.

The analysis of our results indicates that the two methods used in this study are reliable and reproducible for differentiating *P. brasiliensis* and *P. americana*. Moreover, these techniques can be used in a standardized approach for typing *Paracoccidioides* spp. Although sequencing is the gold standard for molecular identification of *Paracoccidioides* spp., the genetic diversity through DNA microsatellites observed between the two species could allow the differentiation of *Paracoccidioides* species by fast and inexpensive methods. Furthermore, these advantages are especially valuable in situations where laboratory facilities are relatively limited. For this, more studies are needed, including a larger number of isolates and all species of the genus *Paracoccidioides*.

## 5. Conclusions

*P. brasiliensis sensu stricto* and *P. americana* have been identified as the species responsible for PCM in the state of Rio de Janeiro. The chronic form of PCM appears to be predominant among patients infected with *P. americana* in this region. Evolutionary analyses, utilizing two different sequencing methods and PCR fingerprint profiles, suggest that a single population of *P. americana* is responsible for the PCM cases reported in Rio de Janeiro and may differ from populations of *P. americana* found in other regions of Brazil. However, given the limited number of *P. americana* strains analyzed in this study, further research with larger sample sizes is needed to infer the phylogeographical patterns and clinical features of the disease caused by both *P. brasiliensis sensu stricto* and *P. americana* in the state of Rio de Janeiro.

## Figures and Tables

**Figure 1 jof-11-00134-f001:**
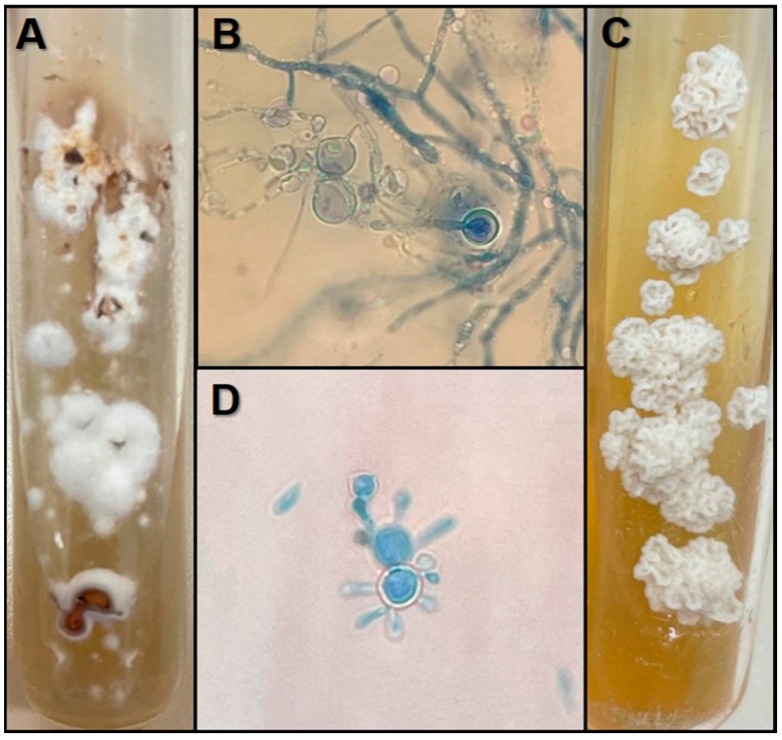
Macromorphology of *Paracoccidioides americana* on Potato Dextrose Agar at 25 °C (**A**); micromorphology of the filamentous form at 25 °C, stained with lactophenol and cotton blue solutions, 40X magnification (**B**); macromorphology of *Paracoccidioides americana* on Fava-Netto Agar at 35 °C (**C**); micromorphology of the yeast-like form at 35 °C, stained with lactophenol and cotton blue solutions, 40X magnification (**D**).

**Figure 2 jof-11-00134-f002:**
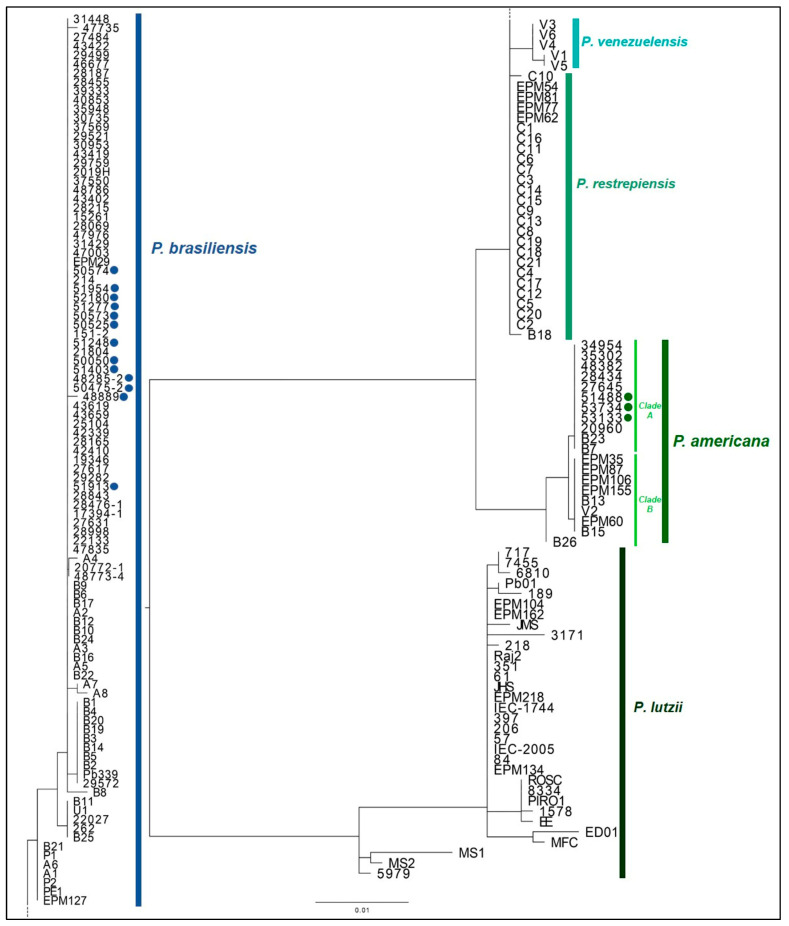
Phylogenetic tree including the strains identified in this study and the reference strains. The relationships of arf and gp43 genes among the 16 clinical isolates included in this study (*P. brasiliensis* marked with dark blue dots; *P. americana* marked with green dots) and 162 strains previously identified are represented. Colors in bars represent the *Paracoccidioides* species described. Dark blue: *P. brasiliensis sensu stricto*; light blue: *P. venezuelensis*; light green: *P. restrepiensis*; green: *P. americana*; dark green: *P. lutzii*. Clades A and B of *P. americana* are represented with lime bars.

**Figure 3 jof-11-00134-f003:**
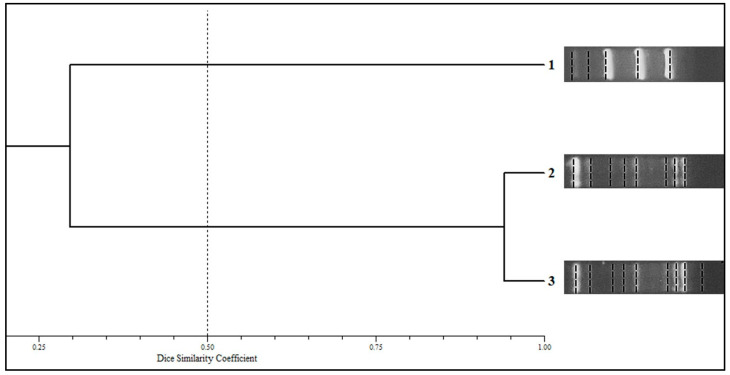
Dendrogram based on the analysis of molecular profiles obtained from PCR products using the primer M13. The similarity between the molecular profiles was analyzed using UPGMA at and NTSYS-pc software (version 2.02h, Applied Biostatistic Inc.) based on the Dice similarity coefficient. Clade 1 (*n* = 10): *P. americana* isolates, including Pb03; Clade 2 (*n* = 11): *P. brasiliensis* isolates, including Pb18; Clade 3 (*n* = 3): *P. brasiliensis* isolates.

**Figure 4 jof-11-00134-f004:**
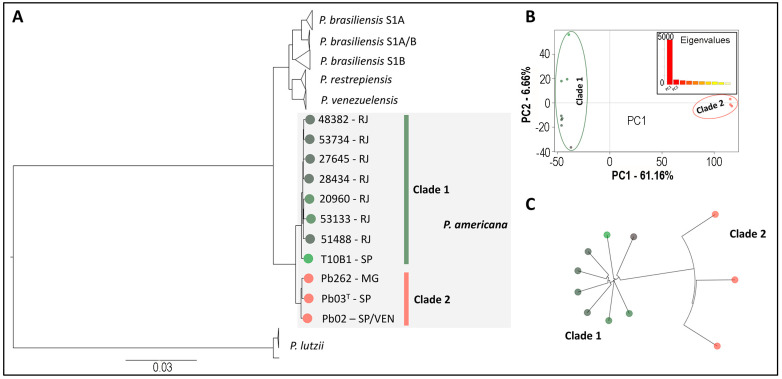
Evolutionary analyses of *Paracoccidioides americana* isolates identified at INI/Fiocruz, Rio de Janeiro, based on whole genome sequencing. (**A**) Maximum likelihood tree based on SNPs collected from 77 genomes of *Paracoccidioides* spp., indicating the clustering of isolates from Rio de Janeiro in the *P. americana* species. Two clades were observed: Clade 1—composed mainly of Rio de Janeiro isolates; and Clade 2—composed mainly of Minas Gerais and São Paulo isolates. (**B**) Principal Component Analysis (PCA) revealed that Clade 1 and 2, in fact, are two distinct populations, as evidenced along the PC1 axis, corresponding to 61.16% of total variation. (**C**) Neighbor Joining Tree unrooted *P. americana* isolates, based on PCA distances, revealing two populations of this species.

**Table 1 jof-11-00134-t001:** Phenotypic and genotypic identification of *Paracoccidioides* spp. isolated from 2015 to 2021 at INI/Fiocruz, Rio de Janeiro, Brazil.

Isolate Code	Species	Date of Fungal Isolation	Probable State of Origin	Clinical Form of PCM
48285-2	*P. brasiliensis*	Dec/15	Rio de Janeiro	Acute
48889	*P. brasiliensis*	June/16	Rio de Janeiro	Chronic
50050	*P. brasiliensis*	Aug/17	Minas Gerais	Chronic
50574	*P. brasiliensis*	Sept/17	Rio de Janeiro	Acute
50475-2	*P. brasiliensis*	Jan/18	Rio de Janeiro	Acute
50525	*P. brasiliensis*	Feb/18	Rio de Janeiro	Chronic
50573	*P. brasiliensis*	March/18	Rio de Janeiro	Acute
51248	*P. brasiliensis*	Sept/18	Rio de Janeiro	Acute
51277	*P. brasiliensis*	Oct/18	Rio de Janeiro	Chronic
51403	*P. brasiliensis*	Nov/18	Rio de Janeiro	Chronic
51488	*P. americana*	Dec/18	Rio de Janeiro	Chronic
51913	*P. brasiliensis*	May/19	Rio de Janeiro	Acute
51954	*P. brasiliensis*	June/19	Rio de Janeiro	Acute
52180	*P. brasiliensis*	Aug/19	Rio de Janeiro	Acute
53133	*P. americana*	Dec/20	Rio de Janeiro	Chronic
53734	*P. americana*	May/21	Rio de Janeiro	Chronic

## Data Availability

The original contributions presented in this study are included in the article. Further inquiries can be directed to the corresponding author.

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
