# Peer review of "Paracoccidioides Species Circulating in the Endemic Area of Rio de Janeiro, Brazil: Updates into Their Genetic Diversity"

_jof, 2025, doi:10.3390/jof11020134_

Round 1

Reviewer 1 Report

The paper titled Paracoccidioides species circulating in the endemic area of Rio de Janeiro, Brazil: updates on their genetic diversity is a contribution to the epidemiological knowledge of  Paracoccidioides in the endemic region. The methods used in this work were adequate to aims proposed. 

Line 127: Phenotypic identification. In this first statement, it is not clear whether the identification was carried out from clinical samples, isolates stored under sterile mineral oil, or isolates grown on PDA.

Line 198. Suggest "Whole genome sequencing was another method performed to evaluate....."

Line 215. Rewrite the sentence

Line 244. Correct the format of the paragraph that remains below the table.

Page 5, 6, 7, 8 and 9. Rewrite de name of fungi in italics.

Author Response

Major comments

The paper titled Paracoccidioides species circulating in the endemic area of Rio de Janeiro, Brazil: updates on their genetic diversity is a contribution to the epidemiological knowledge of  Paracoccidioides in the endemic region. The methods used in this work were adequate to aims proposed. 

Response: Thank you for your comment!

Detail comments

Line 127: Phenotypic identification. In this first statement, it is not clear whether the identification was carried out from clinical samples, isolates stored under sterile mineral oil, or isolates grown on PDA.

Response: Thank you for this suggestion. The section was changed to:

 2.2. Phenotypic identification of isolates grown on PDA and Fava-Netto culture media

All the fungal isolates grown on PDA (Becton, Dickinson and Company, Sparks, MD, USA), and Fava-Netto culture media, with incubation at 25 °C and 37 °C respectively, to obtain their filamentous and yeast forms were identified by their macro and micromorphological characteristics, according to the description made by Lacaz and collaborators [32] – Lines 125-130.

Line 198. Suggest "Whole genome sequencing was another method performed to evaluate....."

Response: Thank you. The whole sentence was changed to:

Whole genome sequencing was another method performed to evaluate the intraspecific genetic variability of P.americana isolates. For this, the DNA from four clinical isolates of P. americana previously identified by our group [30] as well as DNA from P. americana (n=3) isolated from patients with PCM from Rio de Janeiro state and treated at INI/Fiocruz during this study were included in our analysis – Lines 190-194.

Line 215. Rewrite the sentence

Response: Thank you for this suggestion. The sentence was modified to “ In microscopy, large, round, yeast-like cells were observed” – Line 213.

Line 244. Correct the format of the paragraph that remains below the table.

Response: Format corrected. Thank you. (Line 236)

Page 5, 6, 7, 8 and 9. Rewrite de name of fungi in italics.

Response: Sorry for this mistake. All the name of fungi were in italics now.

Reviewer 2 Report

This manuscript describes molecular characterization of a group of Paracoccidioides isolates in Brazil. The results are interesting, but I’m not sure how they compare with previous studies and the conclusions seem broad for the work performed. There are spelling and grammatical errors throughout, which in places caused some confusion about what the authors are trying to say. A lot of this may be clarified with assistance on the English.

Abstract

Lines 21-25: I think I know what you’re getting at, but the writing here is really unclear.

Introduction

Line 42-45: This sentence has a lot of information in it and it’s really long. It may be better to break it up into a couple of sentences.

Lines 50-57: Switching back and forth between acute and chronic makes this hard to follow. I would put all the statements regarding the infection in one place rather than coming back to it mid-way through this paragraph.

Lines 62-63: I’m confused by this. By proven fungal dimorphism, do you mean the transition in the body to the yeast form? Wouldn’t the culture be the thing that proves it?

Lines 66-95: This is really hard to follow and I think repetitive. Is the most relevant information the current species classification? What are the limitations of Sanger sequencing and how does it affect Paracoccidioides?

Line 96: Do the different species affect the methods of diagnosis?

Lines 106-107: I don’t have access to reference 30. How does your study differ from what they did? It would be nice to include a discussion of the findings of the different studies in the results. Were your results confirmatory?

Overall, the introduction could be shortened with only the key points included. If I’m understanding, part of the introduction covers the history of the species identification. Are all of the details here necessary? It seems like the most relevant part is the more recent identification of P. americana.

Materials and Methods

Lines 189-197: The information about the identified isolates belongs in results.

Line 201-202: Wasn’t this how you obtained all of the isolates for this study?

Results

Figure 1 is excellent. Did you grow all 36 isolates or was this limited to the 16 paracoccidioidomycosis strains you identified later?

Lines 230-241: How did you obtain information about acute vs chronic? How certain are you about the origin of the infection?

Lines 285-286: This is stated in the first sentence.

Lines 286-293: It would be helpful to be more clear when you’re talking about the 16 isolates for this study vs the previously identified isolates. Why were those included?

Lines 303-304: More diverse than what?

I may be missing it, but I’m not seeing Figure 4 referenced in the text. Could you clarify what you’re showing here? Are these your results or previous results?

Discussion

Lines 318-328: This is confusing as written. What are the recent epidemiological modifications that make this important to understand? How does it change clinical decisions?

Lines 331-333: With the information you have, how certain are you that these were locally acquired?

Lines 333-334: 2002 or 2016?

Lines 361-365: This seems like overstating based on your data, especially lacking information about the patients (i.e. is it possible that genetic differences of the patients affect the disease form?). I agree further study is warranted.

Lines 366-376: Really hard to follow this. Are there previous studies that suggest that P. americana is less diverse elsewhere? Can you draw this conclusion with 10 isolates?

Conclusion

I would recommend clarifying that these are your results with your isolates and may not be applicable in the entire country. How is the your population different from what is found in other regions? That would need to be explained in the discussion.

Author Response

Major comments

This manuscript describes molecular characterization of a group of Paracoccidioides isolates in Brazil. The results are interesting, but I’m not sure how they compare with previous studies and the conclusions seem broad for the work performed. There are spelling and grammatical errors throughout, which in places caused some confusion about what the authors are trying to say. A lot of this may be clarified with assistance on the English.

Response: Thank you for your comments. We are sure the manuscript is much clear, after the English revision.

Detail comments

Abstract

Lines 21-25: I think I know what you’re getting at, but the writing here is really unclear.

Response: Thank you. The suggested modifications were made as follow: “No intraspecific genetic variation was observed by the M-13 primer in P. americana isolates from this geographic region. However, P. brasiliensis sensu stricto isolates were clustered into two distinct molecular profiles, despite being grouped in a single clade in the phylogenetic tree after partial sequencing of arf and gp43 genes” – Lines 21-24.

Introduction

Line 42-45: This sentence has a lot of information in it and it’s really long. It may be better to break it up into a couple of sentences.

Response: Thank you. The sentence was rewrite to: “The infection is acquired through inhalation of conidia and other infective propagules of the thermodimorphic fungus Paracoccidioides spp. After reach the pulmonary alveoli, they will convert to the yeast phase and are subsequently phagocytosed by cells of the mononuclear phagocytic system with the formation of the primary pulmonary ganglion complex. – Lines 41-45.

Lines 50-57: Switching back and forth between acute and chronic makes this hard to follow. I would put all the statements regarding the infection in one place rather than coming back to it mid-way through this paragraph.

Response: In this paragraph we intend to demonstrate the clinical forms of PCM, since it is a neglected disease and many readers are unaware of its manifestations. Therefore, after mentioning the clinical classification of PCM (lines 47), we briefly describe the characteristics of the infection (line 49) and the disease (line 51). PCM disease can present in two forms, which is why they have been described separately. We hope our collegue agree with us.

Lines 62-63: I’m confused by this. By proven fungal dimorphism, do you mean the transition in the body to the yeast form? Wouldn’t the culture be the thing that proves it?

Response: It would be the transition from the filamentous form to the yeast form. This can be observed by culturing the fungal isolate at room temperature as micelia to yeast cells at a temperature of 37°C. The sentence on Line 60 was rewritten as “The gold standard diagnostic method is culture, and fungal thermal dimorphism is demonstrated through conversion from filamentous to the yeast phase”.

Lines 66-95: This is really hard to follow and I think repetitive. Is the most relevant information the current species classification? What are the limitations of Sanger sequencing and how does it affect Paracoccidioides?

Response: We consider relevant demonstrated the current species classification. Again, the PCM is a neglected disease and the most information we can give to our readers, better will be. With this we summarized the studies on the description of Paracoccidioides species and which are currently considered the causative agents of PCM.

The partial sequencing method may be limited depending on the locus used. For Paracoccidioides, it is important to use more than one locus to differentiate and identify the five species that cause PCM in humans.

Line 96: Do the different species affect the methods of diagnosis?

Response: The different species can affect the serological diagnosis of PCM. Several studies (see references 10; 13; 35) have shown that serum samples from patients infected with one species may not react against antigens obtained from another Paracoccidioides species. Therefore, it is crucial to use antigen preparations that include antigens from the species prevalent in each geographic region or from all species that cause PCM in humans.

Lines 106-107: I don’t have access to reference 30. How does your study differ from what they did? It would be nice to include a discussion of the findings of the different studies in the results. Were your results confirmatory?

Response: Follow the access link to reference 30: https://www.arca.fiocruz.br/handle/icict/64589. I have included it in the references list (Line 508). In the 2018 study (Reference 30), clinical isolates of Paracoccidioides stored in the Collection of Pathogenic Fungi at the Laboratório de Referência Nacional em Micoses Sistêmicas, Instituto Nacional de Infectologia Evandro Chagas, Fundação Oswaldo Cruz (INI/Fiocruz) were identified by molecular biology, in addition to evaluating the clinical, epidemiological, serological, therapeutic, and prognostic aspects of the patients associated with fungal isolates studied. In this study, in addition to the molecular identification of the new clinical Paracoccidioides isolated from PCM patients recently treated at our hospital, we evaluated the intraspecific genetic variability of the identified species (P. brasiliensis sensu stricto and P. americana). In a way, our data are in agreement with the previous study, where in both Paracoccidioides brasiliensis sensu stricto and Paracoccidioides americana (Line 327) were identified.

Overall, the introduction could be shortened with only the key points included. If I’m understanding, part of the introduction covers the history of the species identification. Are all of the details here necessary? It seems like the most relevant part is the more recent identification of P. americana.

Response: Unfortunately, we do not agree with reducing the introduction. As previously mentioned, we believe that the more information we provide about PCM, especially for readers from countries with a low frequency of this mycosis, but who may have cases of the disease through acquisition in endemic countries since the world is more globalized, facilitating its diagnosis, and thus improving its treatment.

Materials and Methods

Lines 189-197: The information about the identified isolates belongs in results.

Response: Thank you for this suggestion. We agree with this comment and made the suggested modifications. The followed sentence was included on the Methodology, 2.5. Intraspecific genetic variability, lines 184-187:  “The DNA from six clinical isolates of P. americana previously identified by our group [30] as well as isolates identified in this study were included in this test. We also have incorporated the DNA from the reference strains Pb03 (P. americana) and Pb18 (P. brasiliensis sensu stricto).”. In the same topic, lines 190-194 the whole sentence was changed to “Whole genome sequencing was another method performed to evaluate the intraspecific genetic variability of P.americana isolates. For this, the DNA from four clinical isolates of P. americana previously identified by our group [30] as well as DNA from P. americana (n=3) isolated from patients with PCM from Rio de Janeiro state and treated at INI/Fiocruz during this study were included in our analysis.”

The previous information was transferred to the Results section, lines 270-272 “After evaluating the 10 samples of P. americana (six previously identified by our group [30]; three identified in this study; and Pb03) and 14 of P. brasiliensis sensu stricto (13 identified in this study and Pb18)”. In the same section, line 293 we wrote: “(four previously identified by our group [30] and three identified in this study)”.

Line 201-202: Wasn’t this how you obtained all of the isolates for this study?

Response: Additional 70 genomes from previous published data were retrieved from SRA database. (See PMID: 27704050 and PMID: 32325168). We added a sentence to clarify this issue in the manuscript (Lines 198-199).

Results

Figure 1 is excellent. Did you grow all 36 isolates or was this limited to the 16 paracoccidioidomycosis strains you identified later?

Response: Thank you for the compliment about Figure 1. The 36 Paracoccidioides isolates were morphologically identified and stored in sterile mineral oil at 25°C by the team at the Laboratório de Referência Nacional em Micoses Sistêmicas, Instituto Nacional de Infectologia Evandro Chagas, Fundação Oswaldo Cruz (INI/Fiocruz). We attempted to recover the 36 isolates on PDA and Fava-Netto culture media, but unfortunately, due this fungus behavior, we were unable to do so. The 16 successfully recovered isolates were identified in this study.

Lines 230-241: How did you obtain information about acute vs chronic? How certain are you about the origin of the infection?

Response: We work at Instituto Nacional de Infectologia Evandro Chagas INI, which comprises a hospital center and 19 laboratories, in addition to all infrastructure such as imaging, nutrition, psychology services, among others. With this, our team is composed by infectologists, dermatologists, and researchers can access the patients’ record system to obtain all the necessary information about, clinical and epidemiological data. In addition, all the patients included in our study have your demographic origin included in their medical records.

Lines 285-286: This is stated in the first sentence.

Response: Second sentence was deleted. Thank you.

Lines 286-293: It would be helpful to be more clear when you’re talking about the 16 isolates for this study vs the previously identified isolates. Why were those included?

Response: The isolates Pb03 and Pb18 were included as they are considered the reference isolates of P. americana and P. brasiliensis sensu stricto, respectively. Since P. americana isolates are rare in this region, we included all those already identified at INI/Fiocruz. The six previously identified (reference 30) and the 3 identified in this study. 

As suggested in the methodology, we included the description: “After evaluating the 10 samples of P. americana (six previously identified by our group [30]; three identified in this study; and Pb03) and 14 of P. brasiliensis sensu stricto (13 identified in this study and Pb18)” – Lines 270-272.

We also rewrote the last paragraph of this result: “Despite this, the clade 2 concentrated most of the P. brasiliensis sensu stricto isolates evaluated (n=11/14; 78.57%), considering ten clinical isolates and Pb18. The acute form of PCM was prevalent in clade 2 (n=8; 80%), while clade 3 grouped only cases of chronic form” – Lines 281-284.

Lines 303-304: More diverse than what?

Response: We agree with this comment, and modifications made. “More diverse than population of others regions” – Results, Line 296.

I may be missing it, but I’m not seeing Figure 4 referenced in the text. Could you clarify what you’re showing here? Are these your results or previous results?

Response: We agree with this comment. We included the reference to Figure 4 in our text in the 3.3.2. Whole genome sequencing (Lines 292-297). The figure shows the two distinct populations of P. americana. These are the results of the WGS conducted in this study.

Discussion

Lines 318-328: This is confusing as written. What are the recent epidemiological modifications that make this important to understand? How does it change clinical decisions?

Response: It was included on the Line 319: “The follow statement “The geographic expansion of PCM cases makes epidemiological surveillance essential to determine which Paracoccidioides species are present in a given region. Furthermore, through epidemiological surveillance, it is possible to use antigen preparations that include antigens from the prevalent species for a more sensitive serological diagnosis. With a more reliable presumptive result, it is possible to initiate early and appropriate treatment for PCM.”

Lines 331-333: With the information you have, how certain are you that these were locally acquired?

Response: Based on the patient's medical history in the medical record, it is possible to suggest the origin of infection. The patient's report provides information about travels, places of residence, profession, and workplaces.

Lines 333-334: 2002 or 2016?

Response: In 2016, an article was published reporting the first case of P. americana in Rio de Janeiro. The clinical isolate was obtained in 2002, however, molecular identification was performed years later in a study carried on by our group.

Lines 361-365: This seems like overstating based on your data, especially lacking information about the patients (i.e. is it possible that genetic differences of the patients affect the disease form?). I agree further study is warranted.

Response: We suggest that the possible genetic modifications undergone by P. brasiliensis over the years may contribute to the different forms of disease presentation.

Lines 366-376: Really hard to follow this. Are there previous studies that suggest that P. americana is less diverse elsewhere? Can you draw this conclusion with 10 isolates?

Response: Yes. Previous reports suggest that P. americana is rare in Brazil (DOI:10.1007/s11046-022-00704-y). The isolates from RJ do not demonstrate intraspecific genetic variability, but they are distinct from isolates from other regions of Brazil. As well as M13 PCR fingerprinting, there are not studies evaluating the intraspecific genetic variability of P. americana isolates through whole genome sequencing analysis.

Conclusion

I would recommend clarifying that these are your results with your isolates and may not be applicable in the entire country. How is the your population different from what is found in other regions? That would need to be explained in the discussion.

Response: Thank you for your recommendation. We revised and rewrote the conclusions (Lines 384-393).

Round 2

Reviewer 2 Report

Thank you for the thoughtful response. There are some spelling/grammatical errors remaining. There is also inconsistency in the use of Paracoccidioidomycosis (species) and P. (species). Lines 80-82 is an example of switching back and forth between the formats.

None.

Author Response

Major comments

Thank you for the thoughtful response. There are some spelling/grammatical errors remaining.

Response: Thank you for this comment. The manuscript was again revised, and we believe the errors were corrected.

There is also inconsistency in the use of Paracoccidioidomycosis (species) and P. (species). Lines 80-82 is an example of switching back and forth between the formats.

Response: According to the taxonomic nomenclature, the first time we cite names of species must be written in full [example: Paracoccidioides restrepiensis (PS3) and Paracoccidioides venezuelensis (PS4)]. In the following quote the name of the Genus can be abbreviated [example: P. restrepiensis].
